# Research on the Control Strategy of Leafy Vegetable Harvester Travel Speed Automatic Control System

Wenming Chen [1,2] , Gongpu Wang [1,*], Lianglong Hu [1], Jianning Yuan [2] , Wen Wu [1,2], Guocheng Bao [1] and Zicheng Yin [1]

1   Nanjing Institute of Agricultural Mechanization, Ministry of Agriculture and Rural Affairs, Nanjing 210014, China
2   Nanjing Institute of Technology, Nanjing 211167, China
*   Correspondence: wanggongpu37@163.com; Tel.: +86-15366092931

**Abstract:** This paper used the 4UM-120D electric leafy vegetable harvester as the research object and designed a travel speed automatic control system to maintain the travel speed within a set value of ±2% in order to improve the efficiency and quality of leafy vegetable harvester operations and decrease the work intensity of the operator. The harvester's travel speed was automatically controlled by using the PID, adaptive fuzzy PID, and sliding mode control techniques after the mechanical and electrical equations for the travel drive motor (a DC brushless motor) were established in MATLAB. By simulating various working situations, the stability, accuracy, and speed of the automatic control system were compared and analyzed using the adjustment time, overshoot, steady-state transition time, and maximum deviation from the set speed as evaluation indicators. The test results revealed that when the current value of the leafy vegetable harvester travel speed deviated from the set value by more than 2%, the dynamic response performance and stability of the DC brushless motor travel drive system based on the sliding mode control strategy was significantly better than that of the PID and adaptive fuzzy PID control strategies, and its anti-disturbance was stronger, achieving the function of automatic control of the harvester travel speed. When the travel motor started with a constant load and the sliding mode control strategy's parameters were the gain factors $A = 1/70$, $c = 100$, $\varepsilon = 100$, and $k = 100$, the travel drive system regulation time was 1.5 s, and the overshoot was 10%. When the harvester was operating smoothly and had leafy vegetable collection baskets loaded and unloaded, the steady-state transition time was 0.3 s. According to the actual engineering application experience, the specific technical state of the control strategy of the agricultural machinery travel speed automatic control system was: regulation time 2.5~3 s; overshoot amount 20~25%; and steady-state transition time 1.0~1.5 s, so the travel speed automatic control system of the electric leafy vegetable harvester in sliding mode was in line with the technical state requirements. The results of the field trials demonstrated the accuracy of the simulation test results. This study offered a method to lessen the work intensity of operators and increase the operating efficiency and quality of a leafy vegetable harvester.

**Keywords:** leafy vegetable harvester; travel speed; automatic control; sliding mode; adaptive fuzzy PID; PID

## 1. Introduction

With a national vegetable-sown area of approximately 21,744,300 hectares and a total production of 782 million tons in 2021, China has the largest variety and widest range of vegetables grown in the world [1,2] (Ministry of Agriculture and Rural Affairs, 2022).

About one-third of the burden in vegetable production operations is made up of labor-intensive and time-intensive harvesting procedures [3,4]. However, manual harvesting still accounts for the majority of domestic vegetable harvesting at the moment,

which severely limits the growth of the vegetable sector [5]. Vegetable harvesting requires mechanization and intelligence more than ever due to the growing labor scarcity in rural areas [6–10]. The actual harvesting procedure revealed various issues and a poor level of intelligent technology despite the fact that a number of vegetable harvesters had recently been placed through promotion trials [11]. With each basket of leafy vegetables collected throughout the harvesting process, the load on the harvester grows. The walking speed will decrease if the power output is not increased. The harvester's walking speed will be slower when it climbs hills or navigates bumps during harvest if the leafy vegetable planting monopoly surface (bed surface) does not fulfill agronomic standards. All of the aforementioned scenarios call for manual speed modification, yet there are drawbacks such as imprecision and huge adjustments being made easily. Inefficient harvesting is caused by excessive travel speed adjustment, which expands the missed cutting zone. However, in the actual harvesting operation, the operator's professionalism is generally lacking, making it challenging to keep the leafy vegetable harvester in a stable working condition for a long time. As a result, the harvester runs too slowly or too quickly, which not only intensifies the operator's work but also has a greater impact on the efficiency and quality of leafy vegetable harvesting [12].

Li Xincheng et al. created a speed detection and control system [13] and used an optical encoder to assess the travel speed of a harvester for leafy vegetables. The benefits and drawbacks of PID and NQL-PID control strategies were discussed by Miao Peng et al., who also created a model of a leafy vegetable harvester walking drive system based on the NQL-PID control algorithm [14]. Rice transplanters with CAN communication interfaces and manual priority were the subject of research by He Jie et al., who also created an expert PID speed-control algorithm for rice transplanters that utilized the Jingguan PZ-60 rice transplanter as a test platform [7]. Guo Hui et al. analyzed the operating principle and control characteristics of a belt-driven stepless variable-speed device, determined the mathematical model of the transmission ratio of the stepless variable-speed device, and designed a wheeled self-propelled square baler travel speed control system based on workload feedback [15].

In this study, the 4UM-120D electric leafy vegetable harvester travel drive system was examined using PID, adaptive fuzzy PID, and sliding mode control techniques. MATLAB was used to construct the related control-strategy model, mechanical equation transfer-function model, and electrical equation transfer-function model for the travel drive motor (DC brushless motor). For simulation studies and field trials, the automatic control system for the harvester's travel speed under the appropriate control strategies was constructed and simulated under various realistic working situations. The stability, accuracy, and speed of the travel speed automatic control system under different control strategies were compared and analyzed using the adjustment time, overshoot, steady-state transition time, and maximum deviation from the set speed as evaluation indicators.

## 2. Materials and Methods

### 2.1. Machine Structure and Working Principle

#### 2.1.1. Machine Structure and Technical Parameters

The 4UM-120D electric leafy vegetable harvester that was used primarily included a cutting mechanism, a reeling mechanism, a cutter height adjustment device, a conveying mechanism, a control box, a 48 V lithium battery, a differential speed, a reducer, a travel drive motor, a control panel, a gear-switching handle, a brake handle, and wheels, among other components. Figure 1 depicts its fundamental layout. The cutter height adjustment tool was composed of a slide rail and an electric push rod. Table 1 lists the machine's primary structural and technical data.

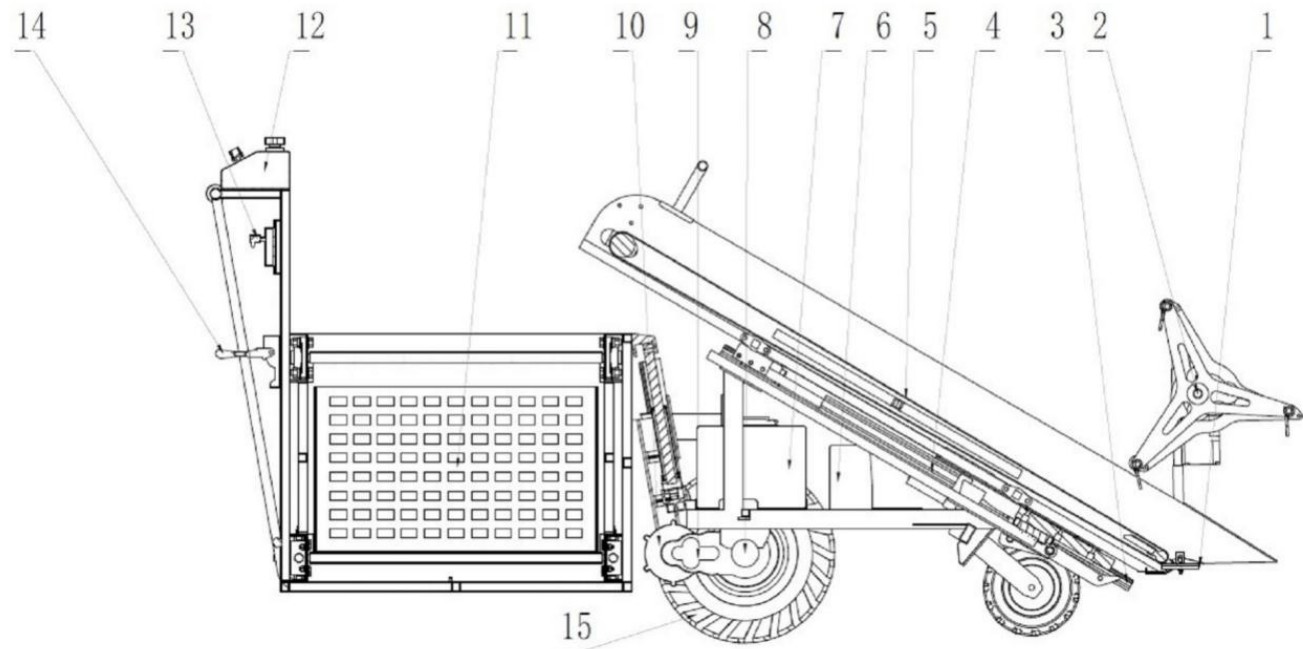

**Figure 1.** Sketch of the structure of the 4UM-120D electric leafy vegetable harvester. 1. Reciprocating double-action cutter; 2. Paddle mechanism; 3. Slide rail; 4. Electric pusher; 5. Conveying mechanism; 6. Control box; 7. 48 V lithium battery; 8. Differential mechanism; 9. Reducer; 10. Travel drive motor; 11. Collection basket; 12. Control panel; 13. Gear switching handle; 14. Brake handle; 15. Wheel.

**Table 1.** The 4UM-120D electric leafy vegetable harvester's structural and technical parameters.

| Parameters | Values |
| --- | --- |
| Whole machine size (length × width × height)/(mm × mm × mm) | 2180 × 1500 × 1200 |
| Battery capacity/ | 50 |
| Working width/ | 1200 |
| Cutter height adjustment range/ | 0~100 |
| Conveyor belt width/ | 1200 |
| Conveyor belt installation inclination/ | 30 |
| Wheel base/ | 550 |
| Wheel radius/ | 175 |
| Minimum ground clearance/ | 70 |
| Productivity/ | 0.04–0.08 |

### 2.1.2. Working Principle

The reciprocating double-action cutter was driven by a DC brushless motor to cut at a set pace when the electric leafy vegetable harvester was in use. The cutter height adjustment device maintained the cutter's height above the ground at an appropriate level for green vegetable stubble. The chopped leafy vegetables were first picked by the paddle wheel and delivered to the conveyor system, where they were then moved to the rear outlet and lastly covered by the collection basket to complete the vegetable collection.

### 2.2. Travel Speed Automatic Control System Components

The 4UM-120D electric leafy vegetable harvester served as the foundation for the travel speed automatic control system, which included a touch screen, PLC, travel motor and its driver, and Hall speed sensor, as illustrated in Figure 2. Figure 3 depicts the system program's flow chart, which was primarily created to fulfill the functions for measuring, displaying, and controlling the walking pace. The PLC calculated the bus voltage value at both ends of the travel motor using various control strategies and the travel motor adjusted the speed according to the size of the bus voltage at both ends to realize the function of

travel speed automatic control. The PLC calculated the difference between the current value of the travel speed and the set value to derive the deviation amount.

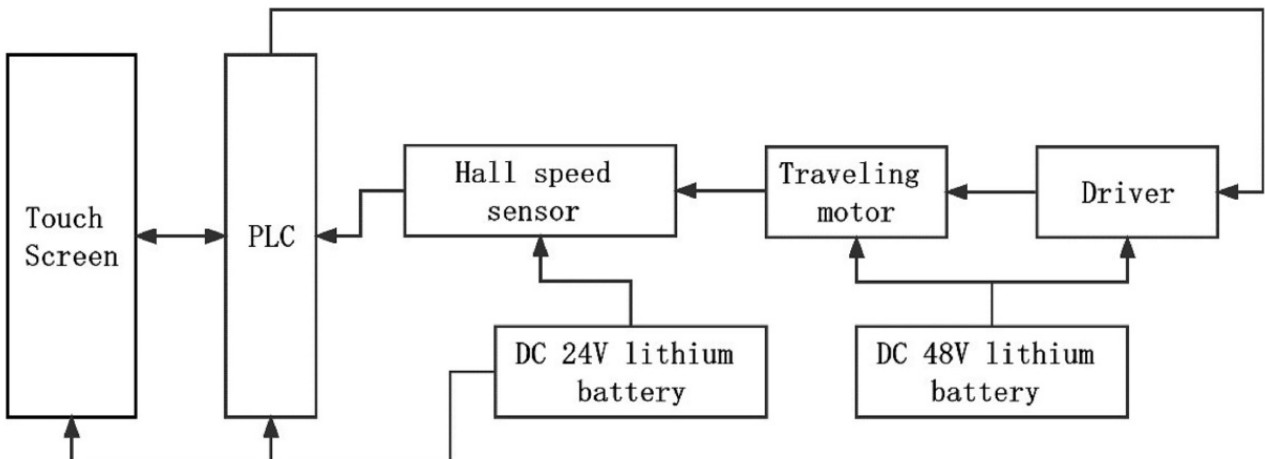

**Figure 2.** Automatic travel speed control system.

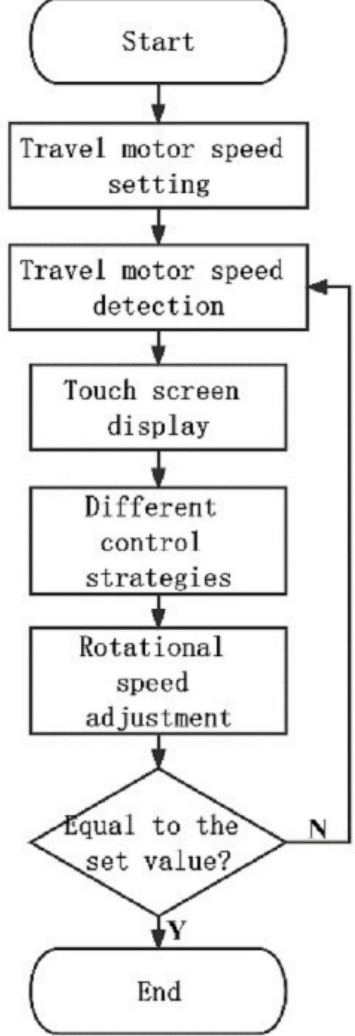

**Figure 3.** Flow chart of the automatic travel speed control program.

*2.3. Model of the Travel Drive System*

2.3.1. Model of the Travel Drive Motor

The 4UM-120 electric leafy vegetable harvester's travel driving system used a DC brushless motor that was modeled to develop a transfer function between its output angular speed and input voltage because the motor input voltage was controlled by adjusting the motor driver output voltage to regulate the motor angular speed. The main stator, main rotor, electronic switching circuit, and position sensor made up the DC brushless motor's four primary components. To simplify the analysis, a three-phase, two-pole permanent magnet DC brushless motor was used as an example, and it was assumed that: (1) the stator winding was 60°, the phase band was concentrated at the whole distance, and the windings were star-connected; (2) the tooth slot effect was neglected and the windings were uniformly distributed on the inner surface of the smooth stator; (3) no armature reaction was considered and the air gap magnetic field distribution was approximately rectangular with a waveform flat top width of 120° electrical angle; and (4) there were no damping windings on the rotor surface, and the permanent magnets were not damped [16]. The motor stator three-phase-winding voltage balance equation is shown in Equation (1), the electromagnetic torque generated by the stator three-phase winding is shown in Equation (2), and the rotor motion equation is shown in Equation (3):

$$
\begin{bmatrix} U_A \\ U_B \\ U_C \end{bmatrix} = \begin{bmatrix} R_A & 0 & 0 \\ 0 & R_B & 0 \\ 0 & 0 & R_C \end{bmatrix} \begin{bmatrix} i_A \\ i_B \\ i_C \end{bmatrix} + \begin{bmatrix} L_A & L_{AB} & L_{AC} \\ L_{BA} & L_B & L_{BC} \\ L_{CA} & L_{CB} & L_C \end{bmatrix} P \begin{bmatrix} i_A \\ i_B \\ i_C \end{bmatrix} + \begin{bmatrix} e_A \\ e_B \\ e_C \end{bmatrix} \tag{1}
$$

$$
T_e = \frac{e_A i_A + e_B i_B + e_C i_C}{\omega} \tag{2}
$$

$$
T_e - T_L - B_v \omega = J \frac{d\omega}{dt} \tag{3}
$$

where $U_A$, $U_B$, and $U_C$—motor stator three-phase winding voltage (V); $R_A$, $R_B$, and $R_C$—motor stator three-phase winding resistance ($\Omega$); $e_A$, $e_B$, and $e_C$—motor stator three-phase winding counter-electromotive force (V); $i_A$, $i_B$, and $i_C$—motor stator three-phase winding current (A); $L_A$, $L_B$, and $L_C$—motor stator three-phase winding self-inductance (H); $L_{AB}$, $L_{AC}$, $L_{BA}$, $L_{BC}$, $L_{CA}$, and $L_{CB}$—mutual inductance between the three phase windings of the motor stator (H); P—differential arithmetic; $T_e$—electromagnetic torque (N·m); $T_L$—load torque (N · m); J—motor rotor inertia (kg·m$^2$); $\omega$—motor angular speed (rad/s); and $B_v$—coefficient of viscous friction (N·m·s).

The amount of the DC bus voltage at the two ends of the DC brushless motor controlled how fast it rotated. In other words, the bus voltage at the motor's two ends served as the input to the DC brushless motor model's transfer function, which produced the motor's angular speed as the output. When taking the three-phase, full-bridge drive, two-by-two conduction method as an example without considering the load first, the transfer function was:

$$
G_1(s) = \frac{\omega(s)}{U_d(s)} = \frac{K_T}{L_A J s^2 + (R_A J + L_A B_v)s + (R_A B_v + K_e K_T)} = \frac{K_T}{R_A B_v + K_e K_T} \frac{\omega_n^2}{(s^2 + 2\varepsilon\omega_n s + \omega_n^2)} \tag{4}
$$

where $G_1(s)$—transfer function between the bus voltage at both ends of the DC brushless motor and the angular speed of the motor; $\omega(s)$—motor angular speed (rad/s); $U_d(s)$—bus voltage at both ends of the motor (V); $K_T$—electromagnetic torque factor (N · m/A); and $K_e$—anti-potential factor (V · s/rad).

### 2.3.2. Drive Train Model

The harvester drive system consisted of the travel drive motor, reducer, wheels, etc. The drive path was travel drive motor–reducer–wheels; the transfer function between the linear speed of the wheels and the speed of the travel motor was:

$$G_2(s) = \frac{v(s)}{n(s)} = \frac{2\pi r}{i} \qquad (5)$$

where $v(s)$—wheel linear speed (m/s); $n(s)$—travel motor rotation speed (r/s); $r$—wheel radius (m); and $i$—reduction ratio.

A model of the harvester travel drive system that was built in Simulink is shown in Figure 4.

### 2.4. Control Strategy Establishment

### 2.4.1. Adaptive Fuzzy PID Control Strategy Establishment

Based on the travel speed variation $e$ and its rate of change $ec$ [17], an adaptive fuzzy PID algorithm was used to regulate the bus voltage at both ends of the DC brushless motor. The basic domain of the travel speed variation was $(-30, 30)$, the fuzzy domain was $(-3, 3)$, and the quantization levels were negative large, negative medium, negative small, zero, positive small, positive medium, and positive large (*NB*, *NM*, *NS*, *ZO*, *PS*, *PM*, and *PB*, respectively), the basic domain of rate of change of which was $(-0.3, 0.3)$, the fuzzy domain was $(-3, 3)$, and quantization levels were negative large, negative medium, negative small, zero, positive small, positive medium, and positive large (*NB*, *NM*, *NS*, *ZO*, *PS*, *PM*, and *PB*, respectively). The fuzzy controller output variable *Kp* had a basic domain of $(-0.3, 0.3)$, a fuzzy domain of $(-3, 3)$, and a quantization level of negative large, negative medium, negative small, zero, positive small, positive medium, and positive large (*NB*, *NM*, *NS*, *ZO*, *PS*, *PM*, and *PB*, respectively). The output variable *Ki* had a basic domain of $(-0.6, 0.6)$, a fuzzy domain of $(-3, 3)$, and a quantization level of negative large, negative medium, negative small, zero, positive small, positive medium, positive large (*NB*, *NM*, *NS*, *ZO*, *PS*, *PM*, and *PB*, respectively). The affiliation function curves corresponding to the input variables $e$ and $ec$ are shown in Figure 5a,b. The affiliation function curves corresponding to the output variables *Kp* and *Ki* are shown in Figure 6a,b.

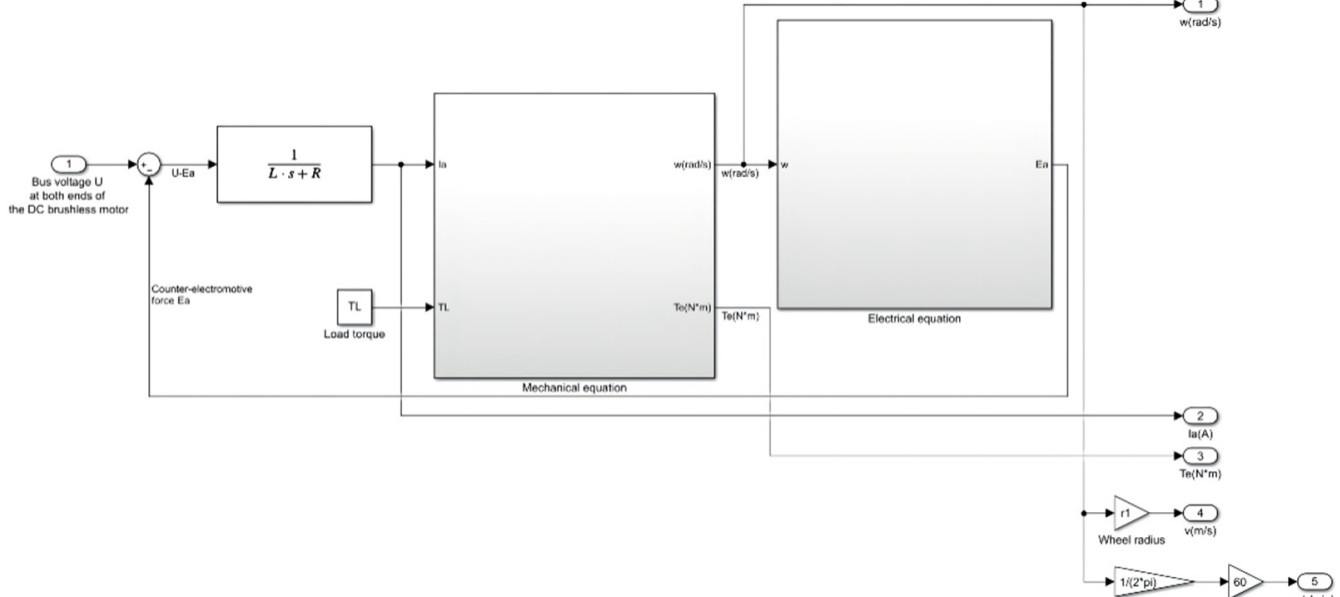

**Figure 4.** Travel drive system model.

It was presumed that *e* was positive when the travel speed was excessively high and negative when it was excessively low. Figures 7 and 8 display the *Kp* and *Ki* rule surfaces. Tables 2 and 3 display the *Kp* and *Ki* fuzzy rule tables [18].

**Table 2.** *Kp* fuzzy control rules table.

| *e* | *ec* | | | | | | |
|---|---|---|---|---|---|---|---|
| | **NB** | **NM** | **NS** | **ZO** | **PS** | **PM** | **PB** |
| NB | PB | PB | PM | PM | PS | PS | ZO |
| NM | PB | PB | PM | PM | PS | ZO | ZO |
| NS | PM | PM | PM | PS | ZO | NS | NM |
| ZO | PM | PS | NS | ZO | NS | NM | NM |
| PS | PS | PS | ZO | NS | NS | NM | NM |
| PM | ZO | ZO | NS | NM | NM | NM | NB |
| PB | ZO | NS | NS | NM | NM | NB | NB |

**Table 3.** *Ki* fuzzy control rules table.

| *e* | *ec* | | | | | | |
|---|---|---|---|---|---|---|---|
| | **NB** | **NM** | **NS** | **ZO** | **PS** | **PM** | **PB** |
| NB | NB | NB | NB | NM | NM | ZO | ZO |
| NM | NB | NB | NM | NM | NS | ZO | ZO |
| NS | NM | NM | NS | NS | ZO | PS | PS |
| ZO | NM | NS | NS | ZO | PS | PS | PM |
| PS | NS | NS | ZO | PS | PS | PM | PM |
| PM | ZO | ZO | PS | PM | PM | PB | PB |
| PB | ZO | ZO | PS | PM | PM | PB | PB |

In the case of a relatively large amount of travel speed variation *e*, when *e* was *PB*, the harvester traveled at an excessively high speed. If at this point *ec* was *PB*, indicating that the tendency for the harvester to continue to travel at a high speed was very high, then the tendency for the travel speed to be low was very high and the angular speed of the DC brushless motor needed to be sufficiently small and the bus voltage at both ends of the DC brushless motor sufficiently reduced (*Kp* was *NB* and *Ki* was *PB*). Conversely, when *e* was *NB*, the travel speed of the harvester was excessively low. If *ec* was *NB* at this point, it showed that the tendency for the harvester to continue to travel at a low speed was very high, so that the tendency to travel at a high speed was very high and the angular speed of the DC brushless motor needed to be sufficiently high and the bus voltage at both ends of the DC brushless motor sufficiently increased (*Kp* was *PB* and *Ki* was *NB*).

In the case of a relatively small amount of travel speed variation *e*, when *e* was *PS*, the harvester traveled at a slightly higher speed. If *ec* was *NS* at this point, indicating a small tendency for the harvester to travel at a low speed, then the travel speed itself slowly decreased at this point and the angular speed of the DC brushless motor decreased gently, so that the bus voltage at both ends of the DC brushless motor itself slowly decreased (*Kp* was *ZO* and *Ki* was *ZO*). Conversely, when *e* was *NS*, the travel speed of the harvester was slightly on the low side. If *ec* was *PS* at this point, indicating a small tendency for the harvester to travel at a greater speed, then the travel speed itself slowly increased, the angular speed of the DC brushless motor increased gently, and thus the bus voltage at both ends of the DC brushless motor slowly increased (*Kp* was *ZO* and *Ki* was *ZO*).

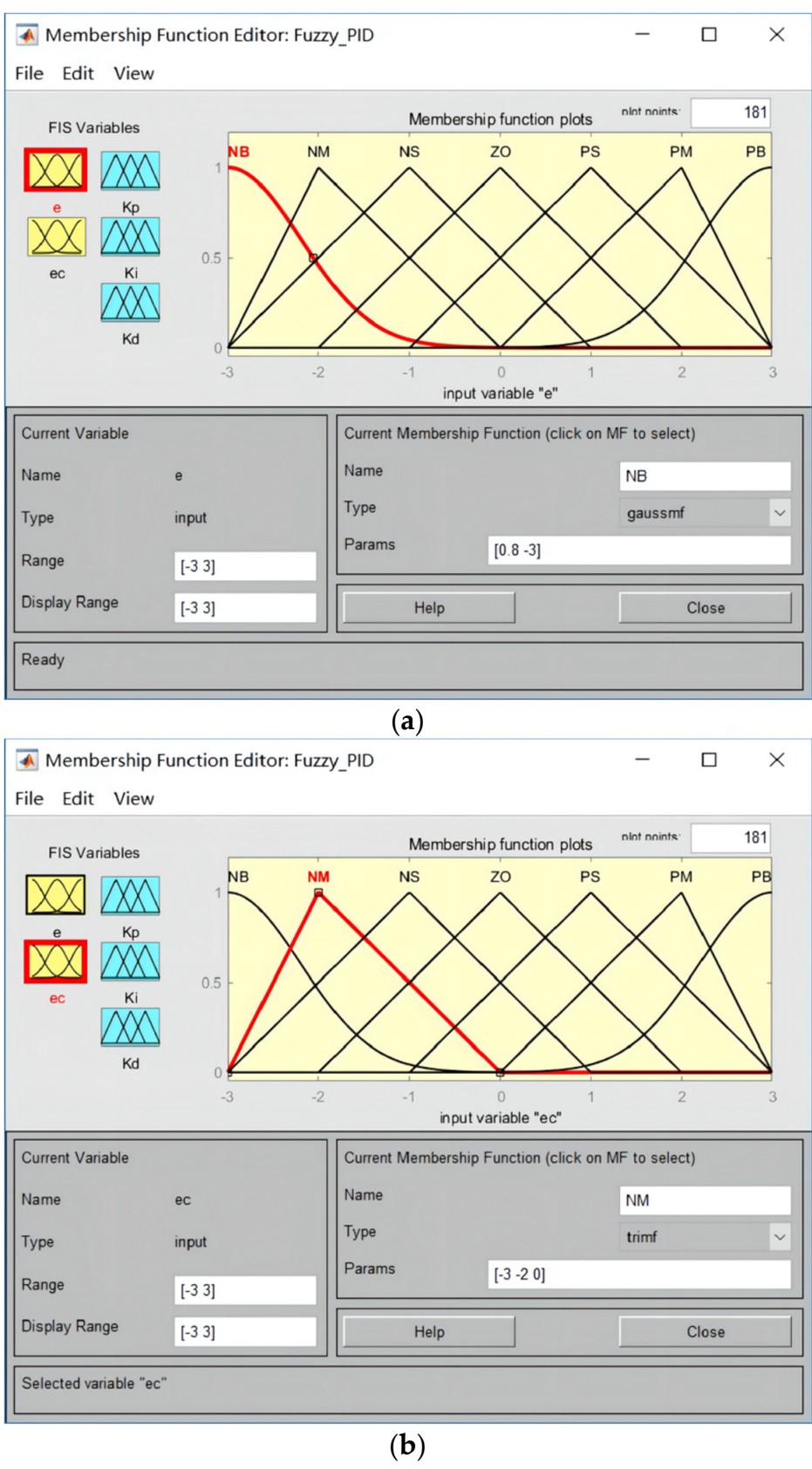

**Figure 5.** (**a**) Affiliation function curve for input variable *e*. (**b**) Affiliation function curve for input variable *ec*.

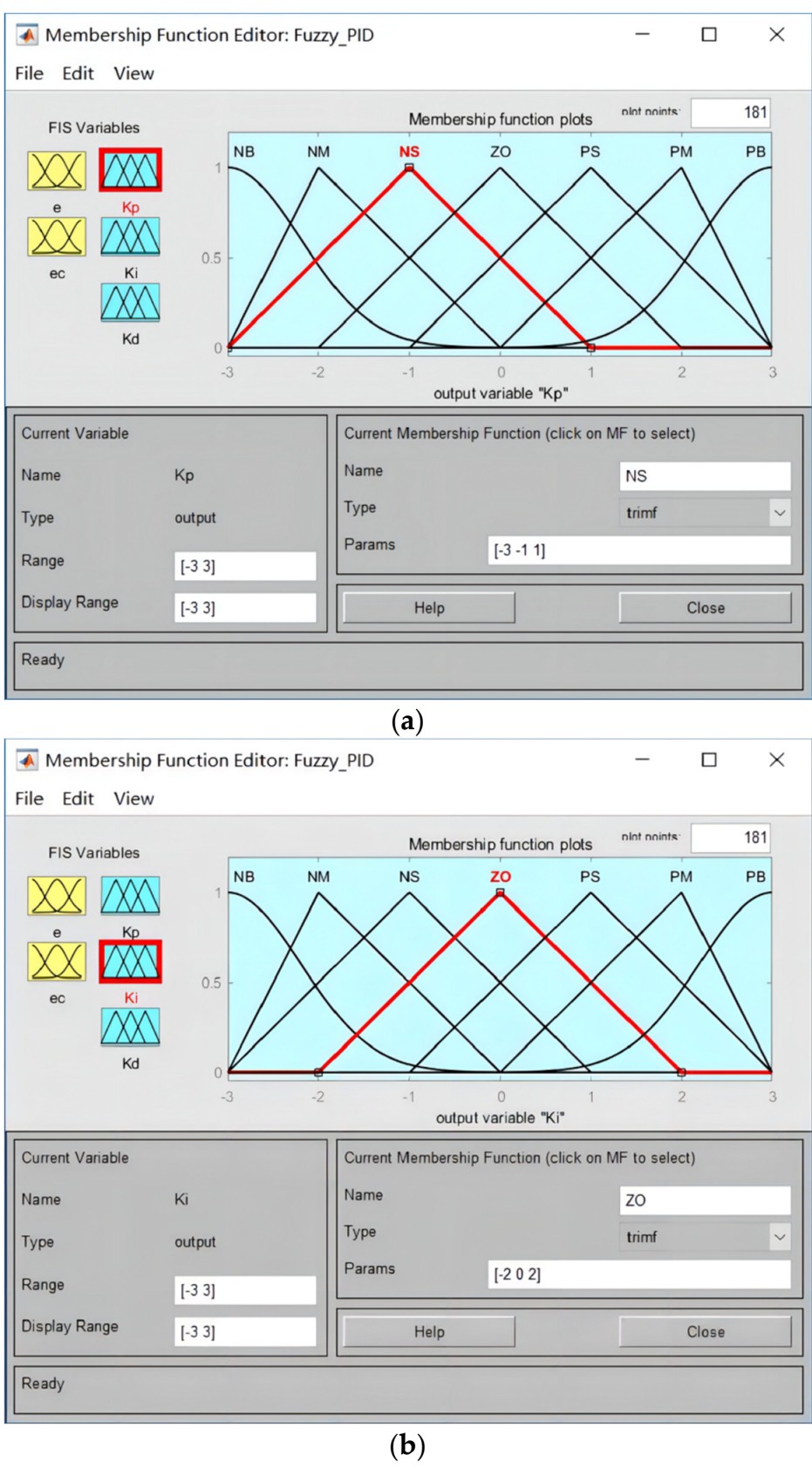

**Figure 6.** (**a**) Affiliation function curve for output variable *Kp*. (**b**) Affiliation function curve for output variable *Ki*.

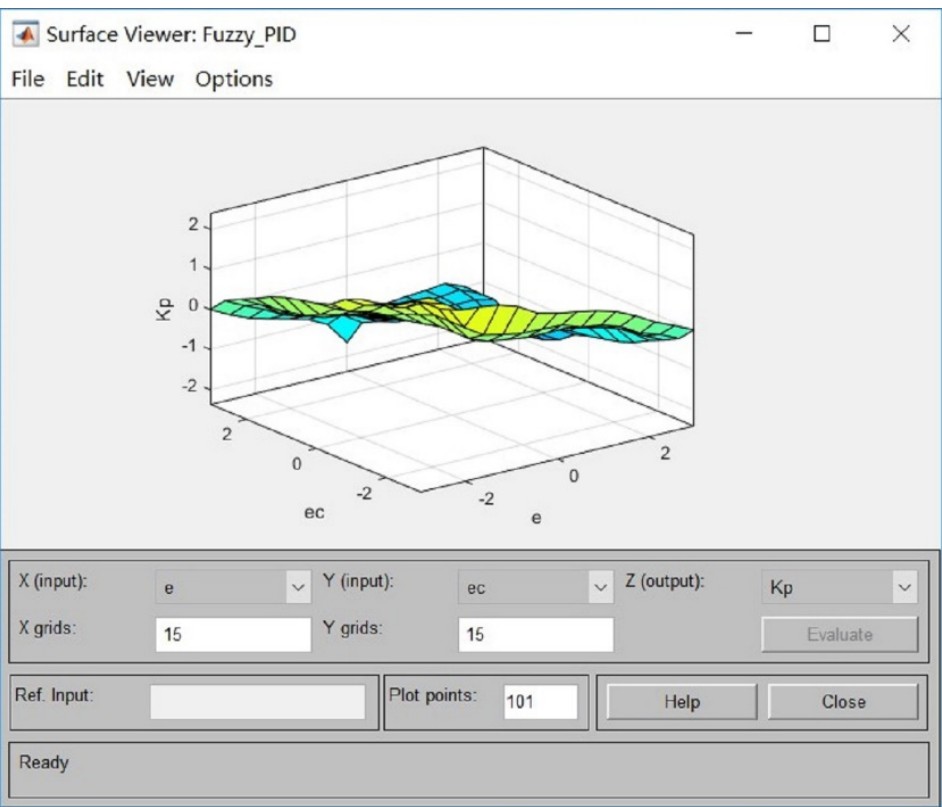

**Figure 7.** *Kp* regular surface.

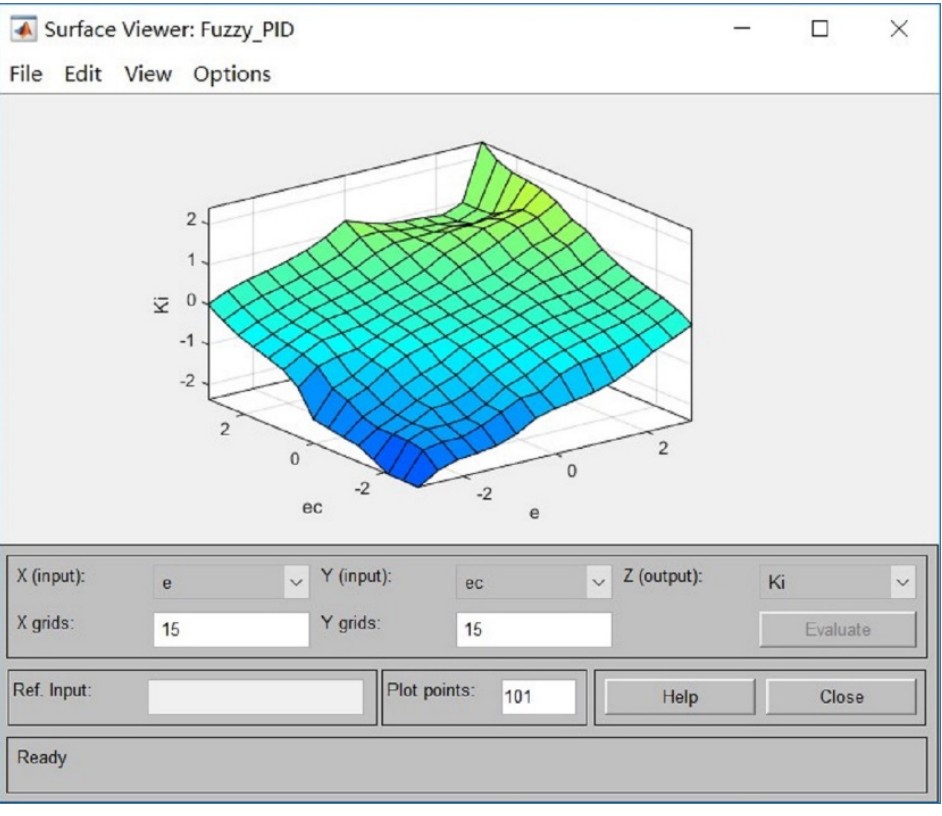

**Figure 8.** *Ki* regular surface.

When *e* was *ZO, ec* was *ZO, Kp* was *ZO,* and *Ki* was *ZO,* the harvester travel speed remained within the set value of $\pm2\%$.

### 2.4.2. Sliding Mode Control Strategy Establishment

We defined the state variables of the leafy vegetable harvester travel drive system as:

$$\begin{cases} x_1 = \omega^* - \omega \\ x_2 = \dot{x}_1 = -\dot{\omega} \end{cases} \tag{6}$$

where $\omega^*$—setting value for angular speed of travel drive motor (rad/s) and $\omega$—current value of the angular speed of the travel drive motor (rad/s).

Combined with the mechanical equation for the DC brushless motor $\frac{T_e - T_L}{J} = \frac{d\omega}{dt}$, we got:

$$\dot{x}_1 = -\dot{\omega} = -\frac{1}{J}(T_e - T_L) = -\frac{1}{J}(K_T I_a - T_L) \tag{7}$$

$$\dot{x}_2 = -\ddot{\omega} = -\frac{K_T}{J}\dot{I}_a \tag{8}$$

where $I_a$—rated current of the travel drive motor (A).

Let $A = \frac{K_T}{J}$, $U = I_a$; then the state space of the leafy vegetable harvester travel drive system was:

$$\begin{pmatrix} \dot{x}_1 \\ \dot{x}_2 \end{pmatrix} = \begin{pmatrix} 0 & 1 \\ 0 & 0 \end{pmatrix} \begin{pmatrix} x_1 \\ x_2 \end{pmatrix} + \begin{pmatrix} 0 \\ -A \end{pmatrix} \dot{U} \tag{9}$$

We designed the sliding modal surface *s* of the travel drive system as:

$$s = cx_1 + x_2 \tag{10}$$

Taking the derivative of *s*, we got:

$$\dot{s} = c\dot{x}_1 + \dot{x}_2 = cx_2 - A\dot{U} \tag{11}$$

The exponential convergence law method could better ensure that the system's point of motion quickly converged to the switching surface while also attenuating the system's sliding mode jitter and made solving for the sliding mode control quantities more straightforward and simple [19] using the following equation:

$$\dot{s} = -\varepsilon sgn(s) - ks \tag{12}$$

where $sgn(s) = \begin{cases} 1, s > 0 \\ -1, s < 0 \end{cases}$, $\varepsilon$, and *k* are both constants greater than zero.

Let $s > 0$ in Equation (12), yielding:

$$\dot{s} = -\varepsilon - ks \tag{13}$$

Solving the differential equation resulted in:

$$s(t) = -\frac{\varepsilon}{k} + \left(s_0 + \frac{\varepsilon}{k}\right)e^{-kt}, s_0 = s(0) \tag{14}$$

Let $s > 0, s(t) = 0$, yielding:

$$\frac{\varepsilon}{k} = \left(s_0 + \frac{\varepsilon}{k}\right)e^{-kt} \tag{15}$$

$$\ln \frac{\varepsilon}{k} - \ln\left(s_0 + \frac{\varepsilon}{k}\right) = -kt \qquad (16)$$

The solution gave:

$$t = \frac{1}{k}\left(\ln\left(s_0 + \frac{\varepsilon}{k}\right) - \ln\frac{\varepsilon}{k}\right) \qquad (17)$$

Thus, the system could reach the switching surface from the initial state in a finite time. The parameter $k$ affected the time it took for the system to reach the switching surface. Increasing $k$ reduced the system regulation time. In order to ensure that the moving point of the system quickly approached the switching surface while weakening the jitter, $k$ had to be increased and $\varepsilon$ had to be reduced at the same time, but too large of a value of $k$ resulted in the moving point approaching the switching surface at too large of a speed; it was not easy to reduce the speed, so it took longer to reach the steady state, so in practical engineering applications, the coefficient $k$ should be combined with the actual system state variables [20].

The exponential convergence law method was used to solve for the sliding mode control quantity $U$ of the leafy vegetable harvester travel drive system, which, when combined with Equations (11) and (12), gave:

$$\dot{s} = -\varepsilon sgn(s) - ks = cx_2 - A\dot{U} \qquad (18)$$

The equation for the sliding mode control quantity $U$ was solved as:

$$U = \frac{1}{A}\int (cx_2 + \varepsilon sgn(s) + ks)dt \qquad (19)$$

In order to verify whether the moving point of the travel drive system was stable after reaching the sliding modal plane, the Liapunov function $V = \frac{1}{2}s^2$ was chosen [21]; according to Liapunov's stability theorem [22], the following conditions needed to be satisfied for the travel drive system to be stable [23,24]:

$$\lim_{s\to 0} s\dot{s} < 0 \text{ and } V \geq 0$$

Clearly, $V = \frac{1}{2}s^2 \geq 0$ satisfied the condition. $\varepsilon$ and $k$ were both constants greater than zero, so $s$ and $\dot{s} = (-\varepsilon sgn(s) - ks)$ were different signs and $\lim_{s\to 0} s\dot{s} < 0$, satisfying the stability theorem and indicating that the walking drive system with the exponential convergence law method sliding mode control strategy was stable.

### 2.5. Control Model Building and Simulation

The PID control strategy model is shown in Figure 9, in which the proportionality factor $KP$ = 13.16 and the integration factor $KI$ = 303.28. The adaptive fuzzy PID control strategy model is shown in Figure 10, where the quantization factor $Ke$ = 10.0, quantization factor $Kec$ = 0.1, proportionality factor $K1$ = 0.1, proportionality factor $K2$ = 0.2, proportionality factor $K3$ = 0, proportionality factor $KP$ = 13.16, integration factor $KI$ = 303.28, and differentiation factor $KD$ = 0.09. The sliding mode control strategy model is shown in Figure 11, where the gain coefficient A = 1/70, the gain coefficient c = 100, the gain coefficient $\varepsilon$ = 100, the gain coefficient = 100, and the gain coefficient k = 100. The model of the travel drive system under the three control strategies is shown in Figure 12.

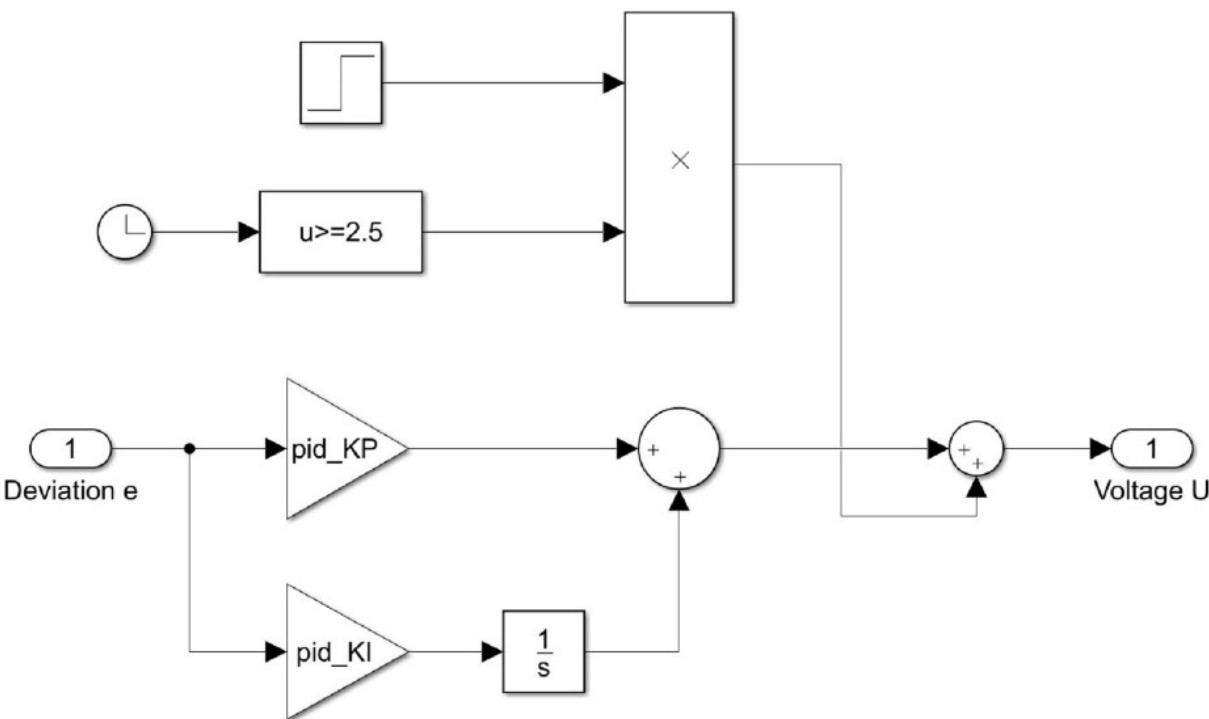

**Figure 9.** PID control strategy Simulink model.

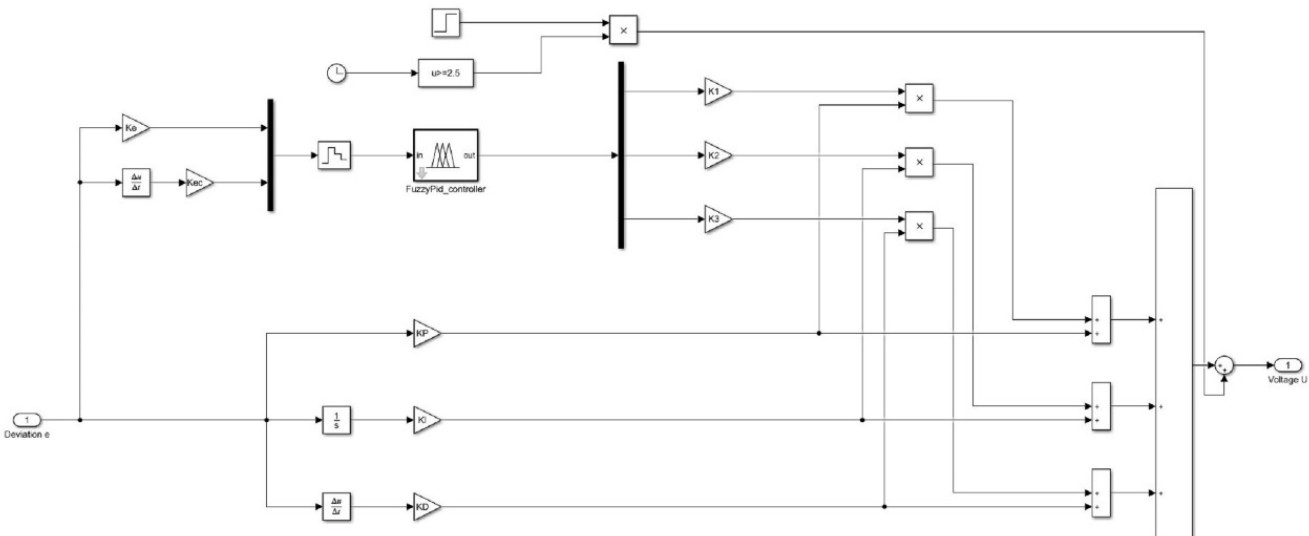

**Figure 10.** Adaptive fuzzy PID control strategy Simulink model.

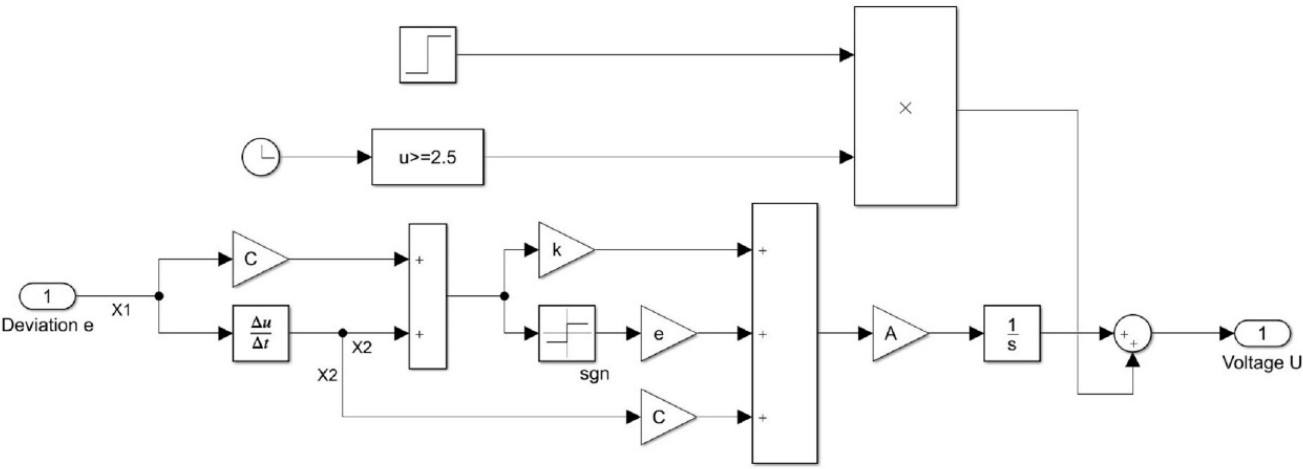

**Figure 11.** Simulink model of sliding mode control strategy.

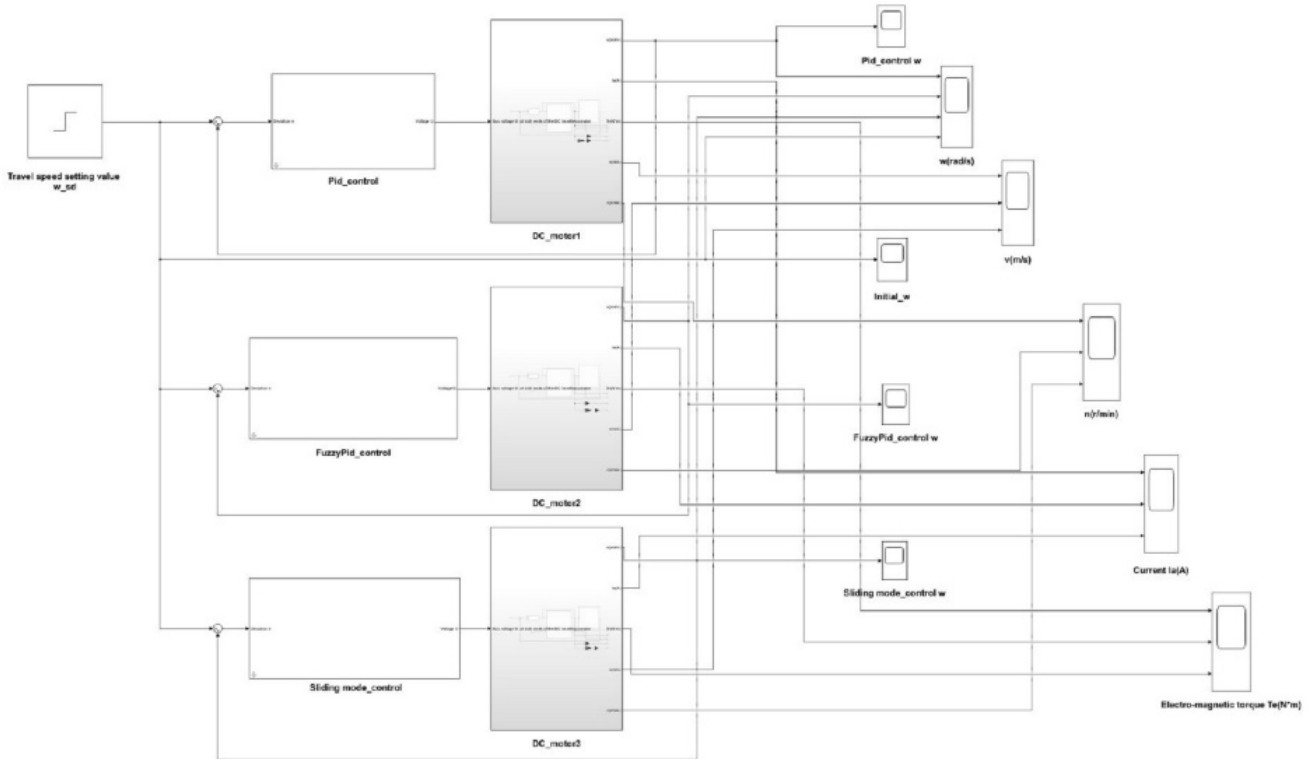

**Figure 12.** Simulink model of a travel drive system with three control strategies.

## 3. Results

### 3.1. Results of the Simulation Tests

According to the above established travel drive system models and control strategy models, the PID-controlled travel drive system model, adaptive fuzzy PID-controlled travel drive system model, and sliding mode controlled travel drive system model were respectively established in MATLAB, including the PID control module, fuzzy control module, PID module, sliding mode control module, travel drive system transfer function, and other modules to collect the angular speed, linear speed, and speed information of the travel drive motor of the leafy vegetable harvester in the harvesting operation in real time. Simulations were carried out for four operating conditions: The simulation test conditions were: constant load starting of the travel motor; sudden load increase under smooth condition: harvester climbing and harvester crossing; and sudden load reduction

under smooth condition: full basket of leafy vegetable unloading machine. The simulation test conditions are shown in Table 4.

**Table 4.** Simulation of experimental conditions.

| Working Condition Numbers | Names | Specific Situations |
| --- | --- | --- |
| Working condition 1 | Constant load starting of trave motor | Constant, unvarying load during start-up of the travel motor |
| Working condition 2 | Sudden increase in load when the travel motor was running smoothly | Harvester climbing suddenly in smooth running operation |
| Working condition 3 | Sudden increase in load when the travel motor was running smoothly | Harvester crossing bump suddenly in smooth running condition |
| Working condition 4 | Sudden decrease in load when the travel motor was running smoothly | Harvester in smooth running condition with leafy vegetable collection baskets filled and unloaded |

(1) Constant load travel motor starting condition: constant and unchanged load during travel motor starting. Simulation of constant load starting of the travel motor: set the angular speed of the travel motor to 2.457 rad/s. The travel motor started at 0.25 s. The load was constant during the starting process. The simulation results are shown in Figure 13a–d. The travel motor angular speed from 0 rad/s always maintained the angular speed set value in a $\pm 2\%$ range of 2.457 rad/s; that is, the travel motor from the start to reaching the stable running state; PID control strategy under the travel drive system adjustment time of 0.3970 s; overshoot of 1.91%; adaptive fuzzy PID control strategy under the system adjustment time of 0.3833 s; overshoot of 1.26%; the system regulation time under sliding mode control strategy was 0.3370 s and the overshoot was 0.45%. Therefore, the dynamic response performance and stability of the travel drive system under the three control strategies were significantly better when the travel motor was started at a constant load, and the sliding mode control strategy was significantly better than the PID and adaptive fuzzy PID control strategies, but the travel drive system that it controlled oscillated slightly in the steady-state range.

(2) The situation 1 of a sudden increase in load when the travel motor was in a smooth state: sudden climbing of the harvester in a smooth running state. The sudden climbing of the harvester in a smooth state: the travel motor had reached a smooth running state 1.5 s ago, and at 1.5 s a slope block obstacle suddenly appeared on the monopoly surface (bed surface). Assuming that the load on the travel motor increased by 40.7% at this time, which was a medium disturbance situation, the simulation results are shown in Figure 14a–d. The travel drive system moved from its original smooth running state to a slope block obstacle and finally backed to a stable state again. The steady-state transition time of the travel drive system under the PID control strategy was 0.3074 s with a maximum deviation from the set speed of 0.0254 rad/s. The steady-state transition time of the system under the adaptive fuzzy PID control strategy was 0.9005 s with a maximum deviation from the set speed of 0.0164 rad/s. The steady-state transition time was 0.0027 s and the maximum deviation from the set speed was 0.0023 rad/s under the sliding mode control strategy.

(3) The situation 2 of a sudden increase in load when the travel motor was in a smooth state: sudden crossing of a bump of the harvester in a smooth running state. The simulation results of the harvester suddenly crossing a bump in a smooth state were as follows: the travel motor reached a smooth running state before 2.0 s, and at 2.0 s, a sudden ditch obstruction appeared on the monopoly surface (bed surface). Assuming that the load on the travel motor increased by 61.1% at this time, which was an oversized disturbance situation, the simulation results are shown in Figure 15a–d. The travel drive system moved from its original smooth running state to a gully obstruction and finally backed to a stable state again. The steady-state transition time of the travel drive system under the PID control strategy was 0.3092 s with a maximum deviation from the set speed of 0.0382 rad/s. The steady-state transition time of the system under the adaptive fuzzy PID control strategy was 0.9621 s with a maximum deviation from the set speed of 0.0247 rad/s. The steady-state transition time was 0.0027 s and the maximum deviation from the set speed was 0.0023 rad/s under the sliding mode control strategy.

Therefore, regardless of the sudden increase in load in the smooth running state of the travel motor, whether it was a sudden slope block obstacle or a ditch obstacle on a monopoly (bed), the travel drive system under the sliding mode control strategy had better immunity to disturbances and stability compared to the PID and adaptive fuzzy PID control strategies, but it oscillated slightly in the range of the secondary steady state.

(4) Sudden load shedding under smooth running of the travel motor: leafy vegetable collection baskets were filled and unloaded under smooth running of the harvester. Simulation of the travel motor with a suddenly reduced load in a smooth state: the travel motor had reached a smooth running state before 2.5 s, and at 2.5 s, the leafy vegetable collection baskets were filled with an unloading machine. Assuming that the load on the travel motor had been reduced by 24.4% at this time, which was a less disturbed situation, the simulation results are shown in Figure 16a–d. The travel drive system went from a smooth running state to a sudden load-shedding situation and then backed to a stable state again. The steady-state transition time of the travel drive system under the PID control strategy was 0.3041 s with a maximum deviation from the set speed of 0.0153 rad/s. The steady-state transition time of the system under the adaptive fuzzy PID control strategy was 0.8189 s with a maximum deviation from the set speed of 0.0099 rad/s, The steady-state transition time was 0.0026 s and the maximum deviation from the set speed was 0.0023 rad/s under the sliding mode control strategy. As a result, the travel drive system under the sliding mode control was extremely insensitive to disturbance and more stable than PID and adaptive fuzzy PID control when the load was suddenly reduced in the smooth running state of the travel motor, but it oscillated weakly in the secondary steady-state range.

In summary, when the current value of the angular speed of the travel drive motor deviated from the set value by more than 2%, the travel drive system adjusted the travel speed through different control strategies to keep it within ±2% of the set value, thus realizing the automatic control function of the travel speed of the harvester. The dynamic response performance and stability of the sliding mode control strategy were significantly better than those of the PID and adaptive fuzzy PID control strategies. The sliding mode control strategy was more resistant to perturbations but oscillated weakly in the range of multiple steady states. The parameters of the sliding mode control strategy were: gain coefficient $A = 1/70$; gain coefficient $c = 100$; gain coefficient $\varepsilon = 100$; and gain coefficient $k = 100$.

*3.2. Results of Field Trials*

To verify the accuracy of the simulation test results, the electric leafy vegetable harvester based on PID, adaptive fuzzy PID, and sliding mode travel speed control modes were respectively applied to the vegetable sweet potato base of the Nanjing Institute of Agricultural Mechanization, Ministry of Agriculture and Rural Affairs. The proportionality factor *KP* of the travel speed PID control algorithm was set to 13.16 and the integration factor *KI* was set to 303.28; the quantization factor *Ke* of the adaptive fuzzy PID control algorithm was set to 10.0, the quantization factor *Kec* was set to 0.1, the proportionality factor *K1* was set to 0.1, the proportionality factor *K2* was set to 0.2, the proportionality factor *KP* was set to 13.16, the integration factor *KI* was set to 303.28, and the differentiation factor *KD* was set to 0.09; the gain factor *A* of the sliding mode control algorithm was set to 1/70, the gain factor *c* was set to 100, the gain factor *ε* was set to 100, and the gain factor *k* was set to 100. The field trials were carried out under two operating conditions: the harvester starting at a constant load and the harvester running smoothly with the leafy vegetable collection baskets filled and unloaded (Figure 17). The results are shown in Figures 18 and 19.

As can be seen in Figure 18, at a constant load start of the harvester, the overshoot was 32% for the PID control algorithm-based walking drive system of the electric leafy vegetable harvester, 22% for the adaptive fuzzy PID control algorithm-based walking drive system, and 10% for the sliding mode control algorithm-based walking drive system. The time when the harvester's travel speed first entered the ±2% range of 0.35 m/s and was not exceeded was taken as the adjustment time. Therefore, the adjustment time of the travel drive system based on the PID control algorithm was 2.2 s, the adjustment time of the travel drive system based on the adaptive fuzzy PID control algorithm was 1.9 s, and the adjustment time of the travel drive system based on the sliding mode control algorithm was 1.5 s. As can be seen in Figure 19, the steady-state transition time of the walking drive system based on the PID control algorithm was 1.0 s, the steady-state transition time of the walking drive system based on the adaptive fuzzy PID control algorithm was 0.5 s, and the steady-state transition time of the walking drive system based on the sliding mode control algorithm was 0.3 s when the leafy vegetable collection baskets were filled and unloaded under the smooth running condition of the harvester. According to the actual engineering application experience, the specific technical state of the control strategy of the agricultural machinery travel speed automatic control system was: regulation time 2.5~3 s; overshoot amount 20%~25%; and steady state transition time 1.0~1.5 s, so the travel speed automatic control system of the electric leafy vegetable harvester in sliding mode was in line with the technical state requirements. In conclusion, the dynamic response performance and stability of the electric leafy vegetable harvester travel drive system based on the sliding mode control strategy were significantly better than those of the PID and adaptive fuzzy PID control strategies, and it was more resistant to disturbances; therefore, the simulation test results were reliable.

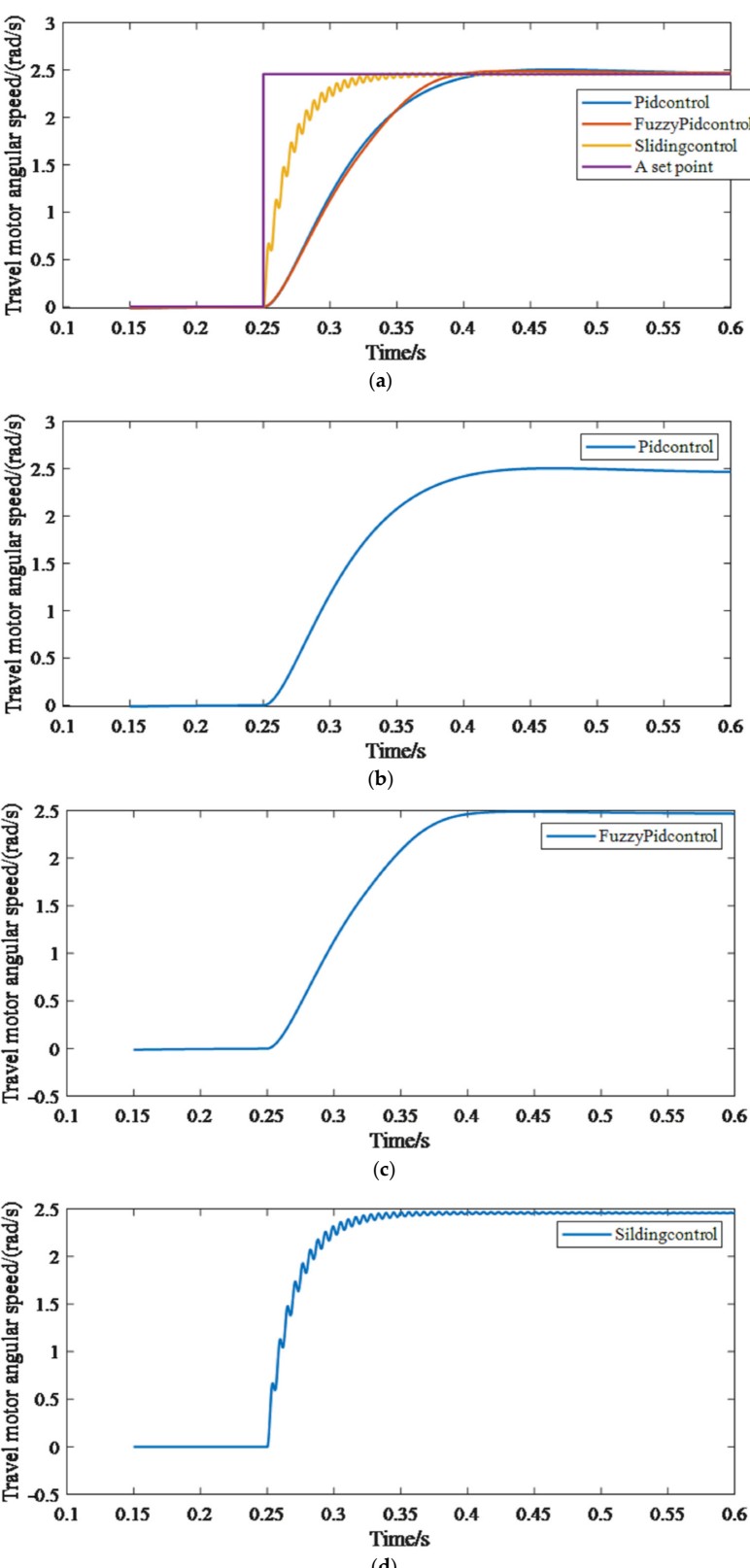

**Figure 13.** (**a**) Simulation results for constant load start of the travel drive system with three control strategies. (**b**) Simulation result for constant load start of the travel drive system with PID control strategy. (**c**) Simulation result for constant load start of the travel drive system with adaptive fuzzy PID control strategy. (**d**) Simulation result for constant load start of the travel drive system with sliding mode control strategy.

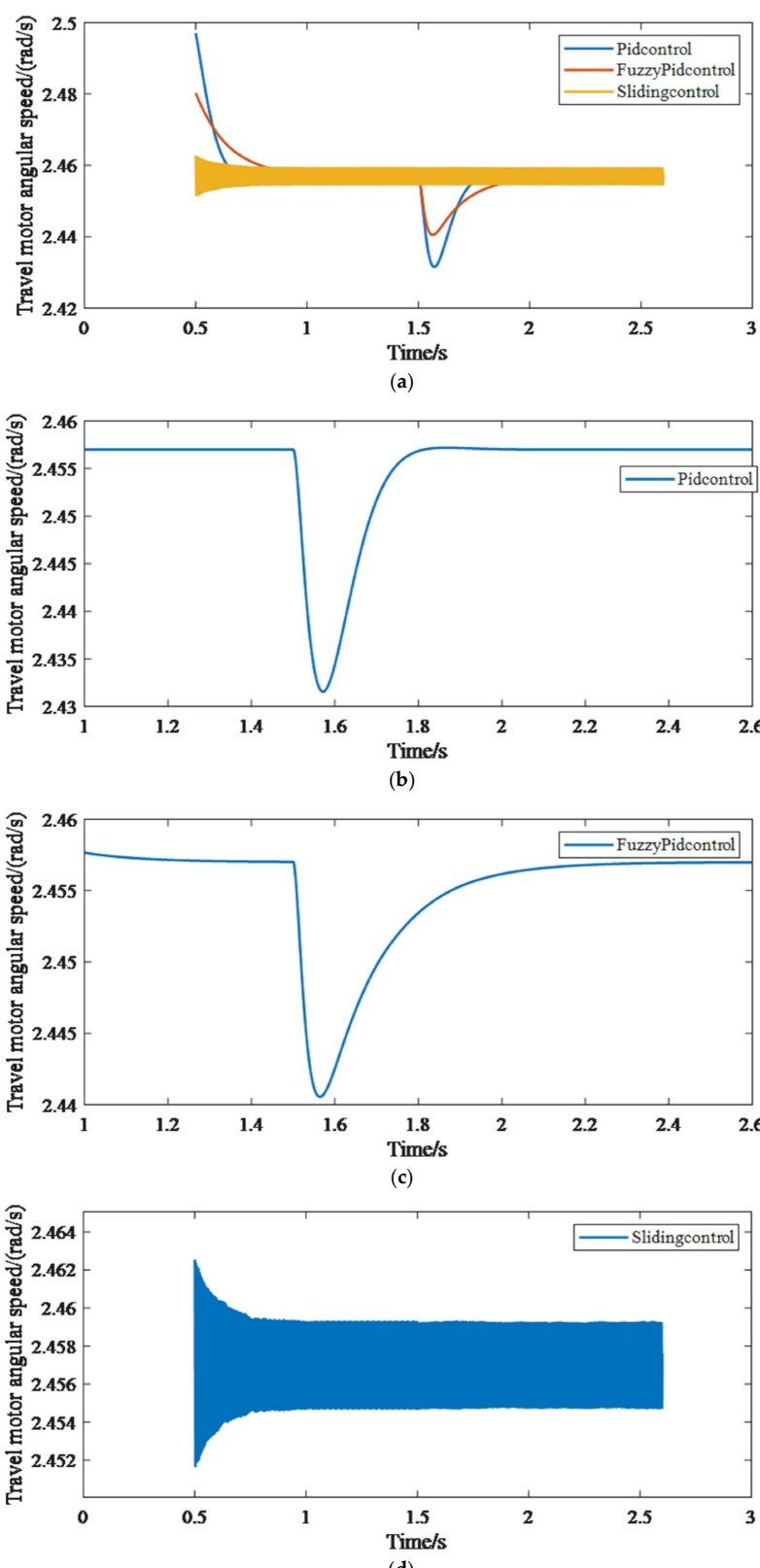

**Figure 14.** (**a**) Simulation results of sudden climbing under smooth conditions of the travel drive system with three control strategies. (**b**) Simulation result of sudden climbing under smooth conditions of the travel drive system with PID control strategy. (**c**) Simulation result of sudden climbing under smooth conditions of the travel drive system with adaptive fuzzy PID control strategy. (**d**) Simulation result of sudden climbing under smooth conditions of the travel drive system with sliding mode control strategy.

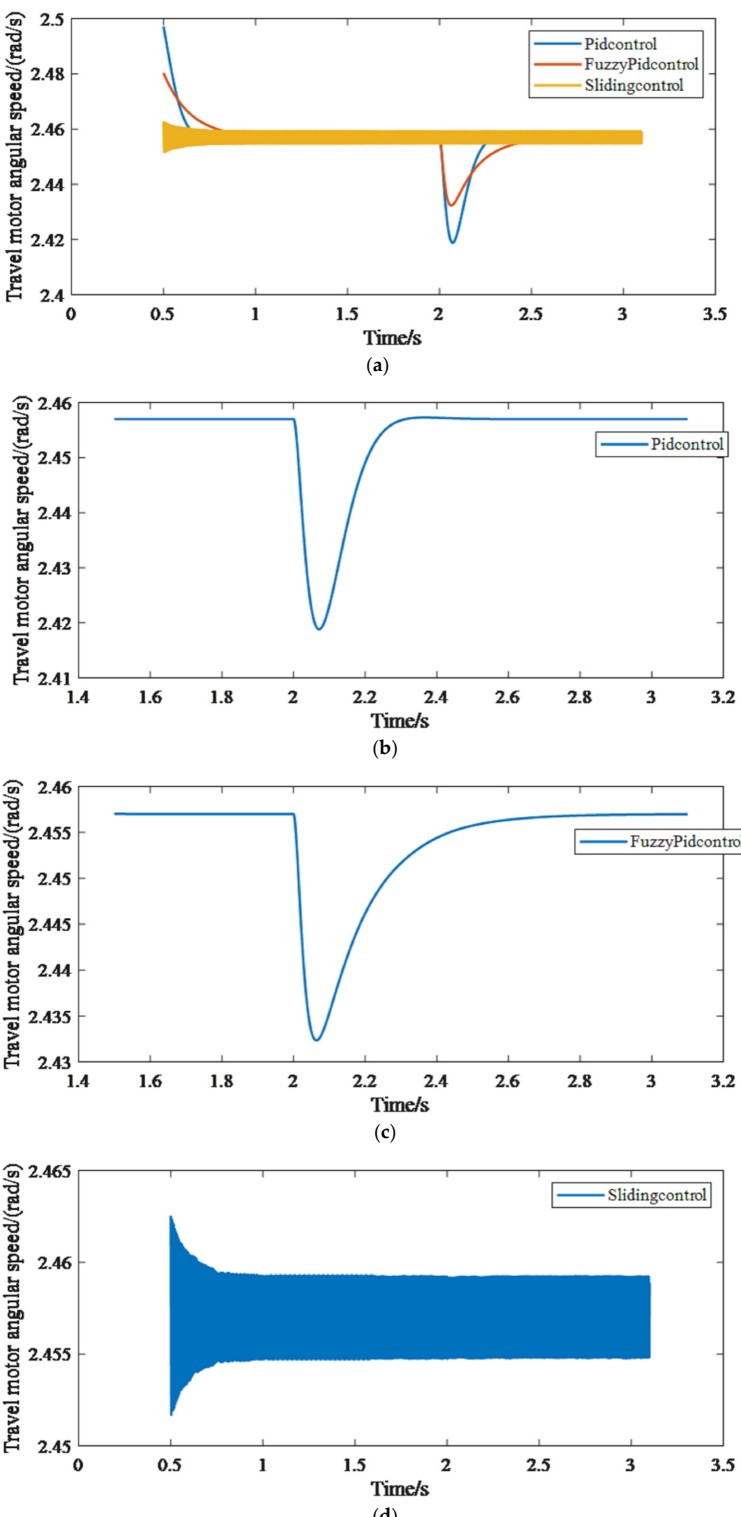

**Figure 15.** (**a**) Simulation results of the sudden crossing of the smooth state of the travel drive system with three control strategies. (**b**) Simulation result of the sudden crossing of the smooth state of the travel drive system with PID control strategy. (**c**) Simulation result of the sudden crossing of the smooth state of the travel drive system with adaptive fuzzy PID control strategy. (**d**) Simulation result of the sudden crossing of the smooth state of the travel drive system with sliding mode control strategy.

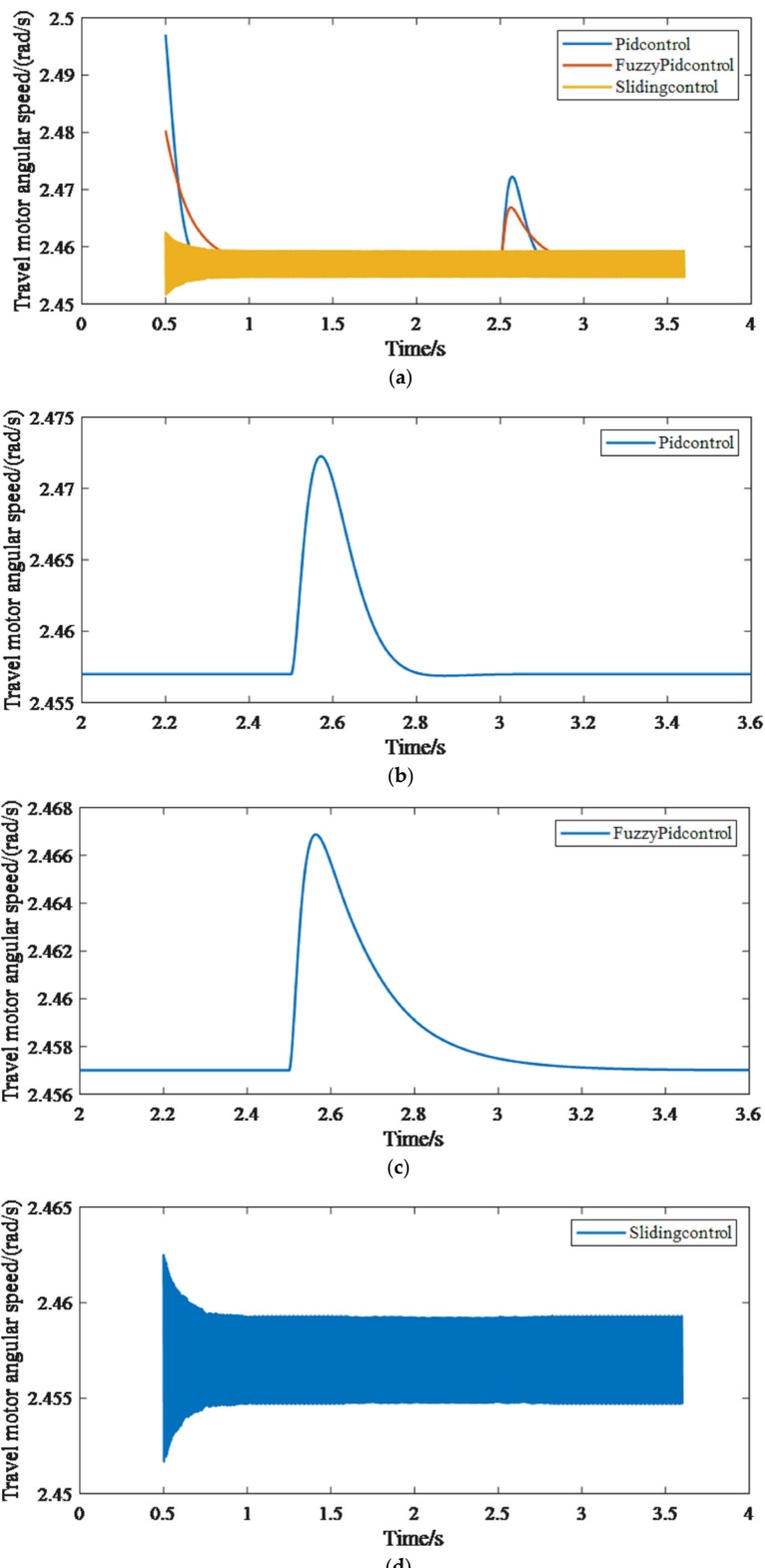

**Figure 16.** (**a**) Simulation results of sudden load shedding under smooth conditions of the travel drive system with three control strategies. (**b**) Simulation result of sudden load shedding under smooth conditions of the travel drive system with PID control strategy. (**c**) Simulation result of sudden load shedding under smooth conditions of the travel drive system with adaptive fuzzy PID control strategy. (**d**) Simulation result of sudden load shedding under smooth conditions of the travel drive system with sliding mode control strategy.

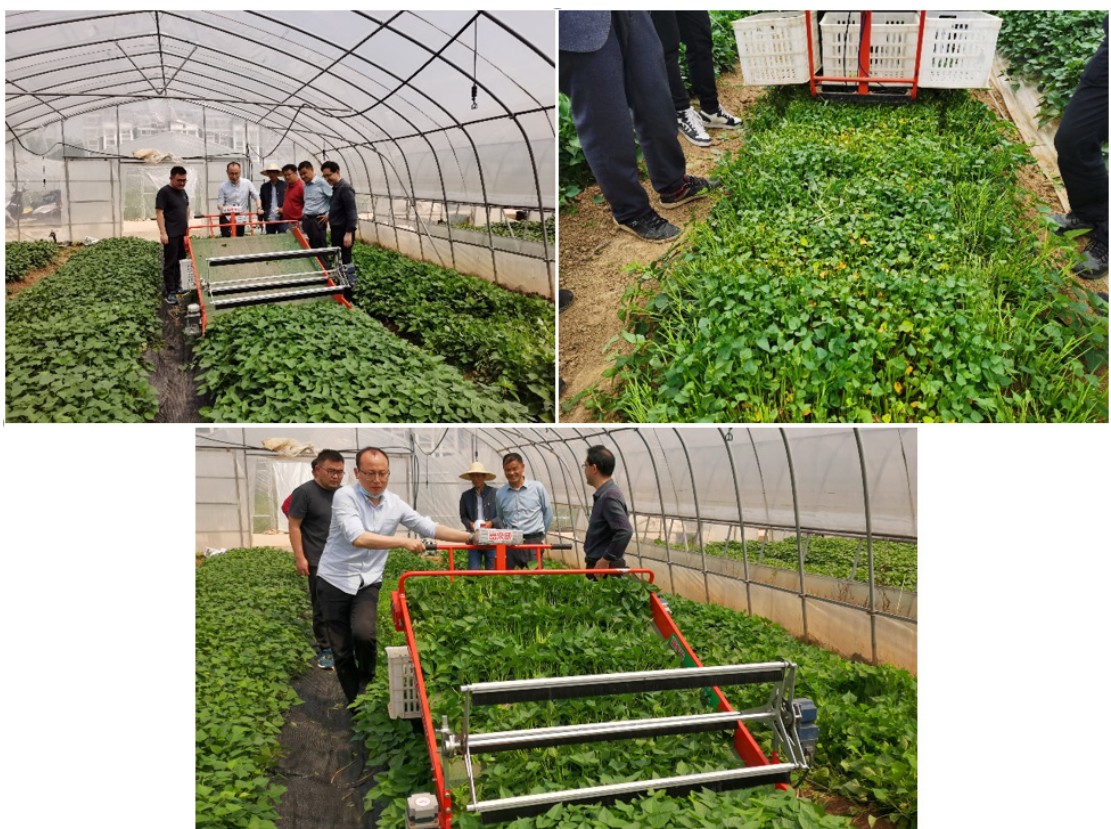

**Figure 17.** Photographs of the 4UM-120D electric leafy vegetable harvester in the field.

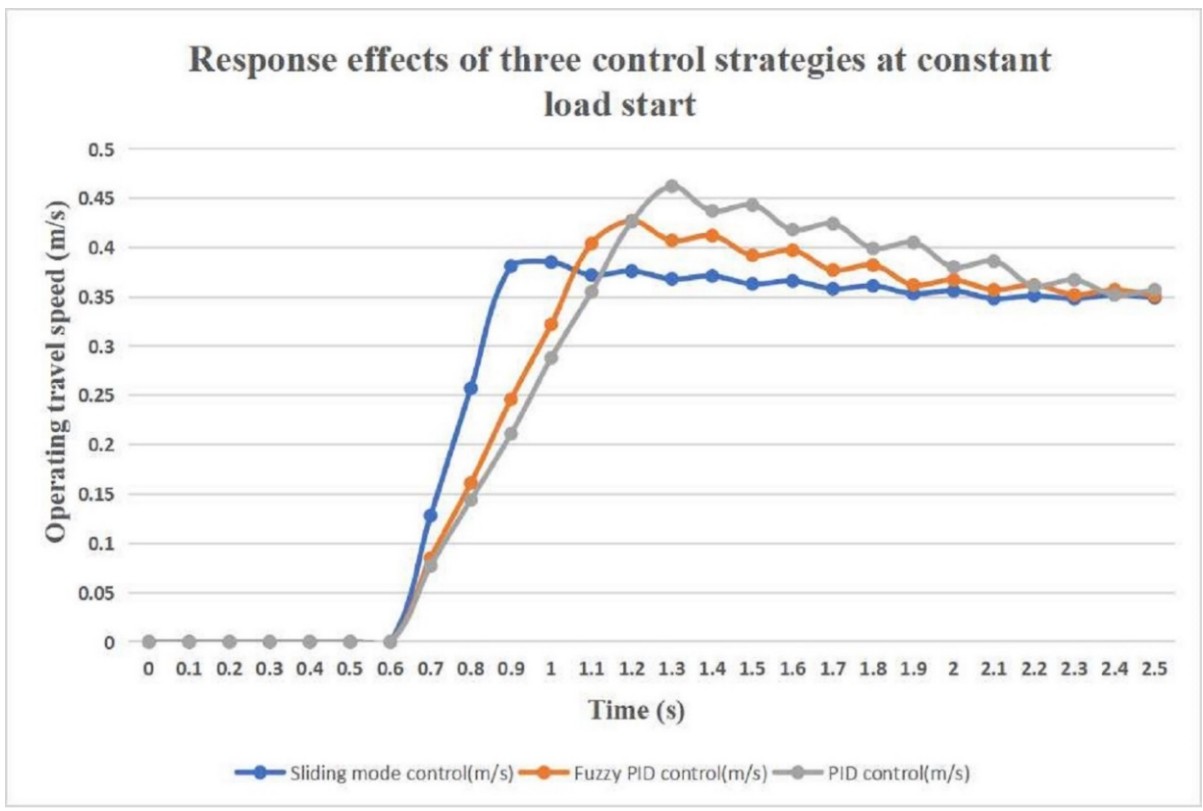

**Figure 18.** Response effects of three control strategies for constant load starting of harvester.

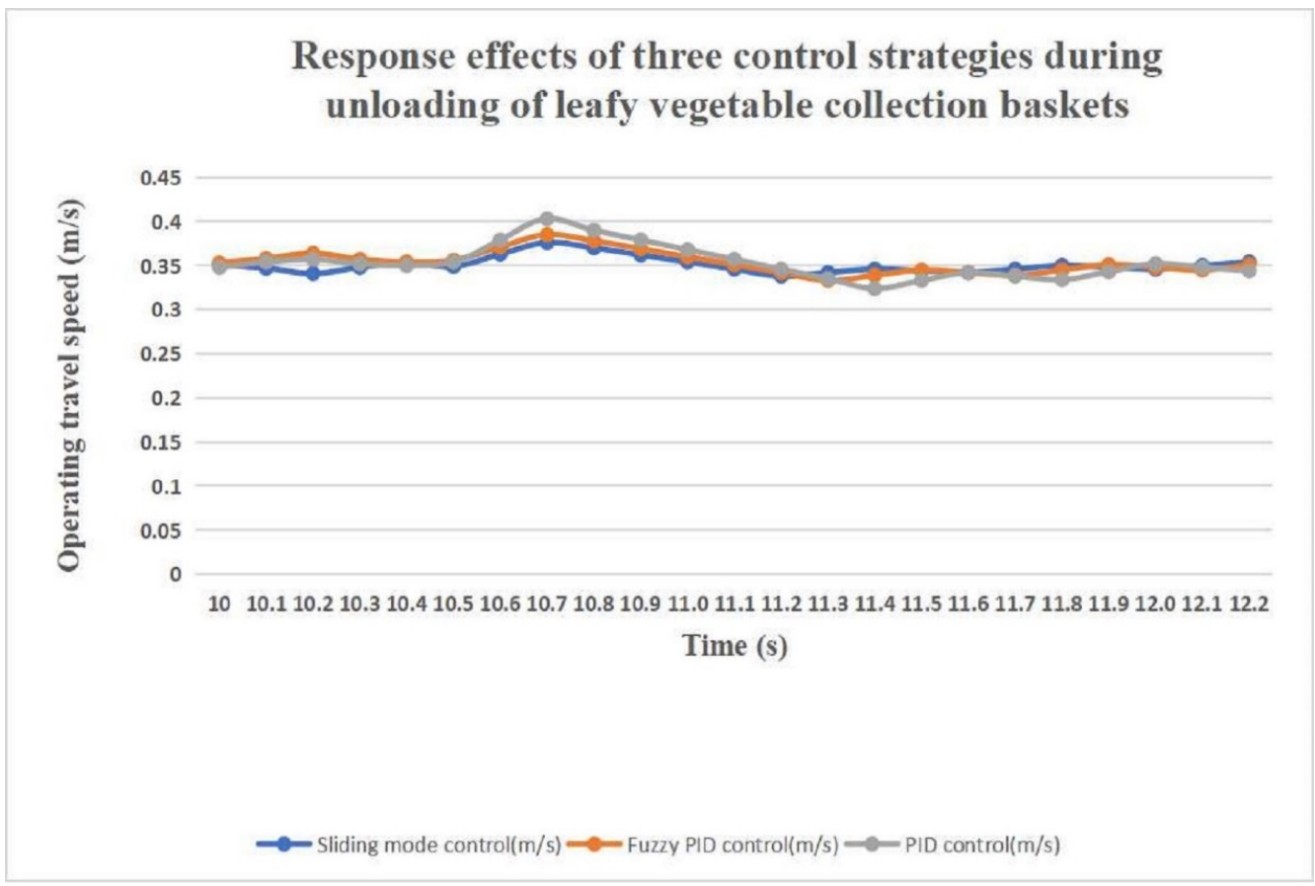

**Figure 19.** Response effects of three control strategies when the leafy vegetable collection baskets were full and unloaded under smooth operation of the harvester.

## 4. Conclusions

(1) In order to achieve automatic control of the travel speed of the electric leafy vegetable harvester, this paper proposed various control strategies. It also established a mathematical model of the travel drive motor based on the physical properties of the DC brushless motor and designed a travel drive system using PID, adaptive fuzzy PID, and sliding mode control techniques.

(2) Through theoretical analysis and simulation analysis of different working conditions and validation of the results of the simulation analysis through field trials, it was shown that the dynamic response performance and stability of the DC brushless motor travel drive system based on the sliding mode control strategy were significantly better than the PID and adaptive fuzzy PID control strategies when the current value of the leafy vegetable harvester's travel speed deviated from the set value by more than 2%, and it was also more resistant to disturbances and achieved automatic control of the harvester's travel speed, resulting in a much higher quality and efficiency of leafy vegetable harvesting while greatly reducing manual labor intensity.

(3) When each parameter of the sliding mode control strategy was: gain coefficient A = 1/70; gain coefficient c = 100; gain coefficient $\varepsilon$ = 100; and gain coefficient k = 100, if the travel motor was started with a constant load, the travel drive system had a regulation time of 1.5 s and an overshoot of 10%; if the harvester was running smoothly with the leafy vegetable collection baskets filled and unloaded, the steady-state transition time of the travel drive system was 0.3 s. According to the actual engineering application experience, the specific technical state of the control strategy of the agricultural machinery travel speed automatic control system was: regulation time 2.5~3 s; overshoot amount 20%~25%; and steady state transition time 1.0~1.5 s, so the travel

speed automatic control system of the electric leafy vegetable harvester in sliding mode was in line with the technical state requirements.

**Author Contributions:** Conceptualization, L.H. and G.W.; methodology, J.Y.; software, W.C.; validation, L.H., G.W. and W.C.; formal analysis, W.C.; investigation, W.W. and Z.Y.; resources, G.B.; data curation, W.C.; writing—original draft preparation, W.C.; writing—review and editing, G.W.; visualization, L.H.; supervision, G.W.; project administration, L.H.; funding acquisition, G.W. All authors have read and agreed to the published version of the manuscript.

**Funding:** This research was supported by the earmarked fund for CARS-10-Sweetpotato (funding number: CARS-10, funder: Lianglong Hu). This research was also funded by the National Key Research and Development Program of China (funding number: 2020YFD1000802-05, funder: Gongpu Wang) and the Key R&D Program of Jiangsu Province: The Key R&D technologies for efficient green production of sweet potato (funding number: BE2021311, funder: Gongpu Wang).

**Data Availability Statement:** Not applicable.

**Conflicts of Interest:** The authors declare no conflict of interest.

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
