# Peer review of "Research on the Control Strategy of Leafy Vegetable Harvester Travel Speed Automatic Control System"

_agriengineering, doi:10.3390/agriengineering4040052_

Round 1
Reviewer 1 Report
1. The advantage of the research is the sliding mode, but the state of the art regarding the sliding mode does not yet exist. the novelty of this research is not clear.
2. Keywords can be more concise but still clear.
3. In equation (1), what is P? Matrix or not? if matrix, what is the order?.
4. Fig. 4 is not clear.
5. Fig. 5 and Fig. 6, Fig. 13 to Fig. 16 are not proportionally located
Author Response
Response to Reviewer 1 Comments
Point 1: The advantage of the research is the sliding mode, but the state of the art regarding the sliding mode does not yet exist. the novelty of this research is not clear.
Response 1: Thank you for your valuable suggestions. I apologize for only giving the evaluation indicators for the sliding mode in the paper, but not the specific state of the art for the sliding mode. I have reviewed the literature on agricultural machinery travel speed automatic control system and found that there is the adjustment time, overshoot amount, steady-state transition time and the maximum deviation from the set speed as the evaluation index of the system's merit, but no specific technical state of the system control strategy was found. According to the actual engineering application experience, the specific technical state of the control strategy of the agricultural machinery travel speed automatic control system is: regulation time 2.5~3s; overshoot amount 20%~25%; steady state transition time 1.0~1.5s, so the travel speed automatic control system of the electric leafy vegetable harvester in sliding mode is in line with the technical state requirements. I am very sorry that I have not expressed the novelty of this study clearly. The novelty of this research lies in the fact that, compared to the PID and fuzzy PID algorithms for the travel speed automatic control system of electric leafy vegetable harvesters, the sliding mode algorithm travel speed automatic control system can enter the set travel speed steady state range much faster and with less fluctuation, resulting in a much higher quality and efficiency of leafy vegetable harvesting, while greatly reducing manual labor intensity.
Point 2: Keywords can be more concise but still clear.
Response 2: Thank you for the Keywords suggested. The precedent version of the Keywords has been replaced, becoming Leafy vegetable harvester; Travel speed; Automatic control; Sliding mode; Fuzzy PID; PID.
Point 3: In equation (1), what is P? Matrix or not? if matrix, what is the order?.
Response 3: Thank you for the P in equation (1) suggested. P = dU/dt in equation (1), which is not a matrix, represents the differential arithmetic.
Point 4: Fig. 4 is not clear.
Response 4: Thank you for the Fig. 4 suggested. The precedent version of the Fig. 4 has been replaced, becoming the Figure 4(a).
Figure 4(a). Travel drive system model.
Point 5: Fig. 5 and Fig. 6, Fig. 13 to Fig. 16 are not proportionally located.
Response 5: Thank you for the Fig. 5 and Fig. 6, Fig. 13 to Fig. 16 suggested. The precedent version of the Fig. 5 and Fig. 6, Fig. 13 to Fig. 16 has been replaced. The Fig. 5 becomes the Figure 5(a) and Figure 5(b). The Fig. 6 becomes the Figure 6(a) and Figure 6(b). The Fig. 13 becomes the Figure 13(a), Figure 13(b), Figure 13(c) and Figure 13(d). The Fig. 14 becomes the Figure 14(a), Figure 14(b), Figure 14(c) and Figure 14(d). The Fig. 15 becomes the Figure 15(a), Figure 15(b), Figure 15(c) and Figure 15(d). The Fig. 16 becomes the Figure 16(a), Figure 16(b), Figure 16(c) and Figure 16(d).
Figure 5(a). Affiliation function curve for input variable e.
Figure 5(b). Affiliation function curve for input variable ec.
Figure 6(a). Affiliation function curve for output variable Kp.
Figure 6(b). Affiliation function curve for output variable Ki.
Figure 13(a). Simulation results for constant load start of the travel drive system with three control strategies.
Figure 13(b). Simulation result for constant load start of the travel drive system with PID control strategy.
Figure 13(c). Simulation result for constant load start of the travel drive system with fuzzy PID control strategy.
Figure 13(d). Simulation result for constant load start of the travel drive system with sliding mode control strategy.
Figure 14(a). Simulation results of sudden climbing under smooth conditions of the travel drive system with three control strategies.
Figure 14(b). Simulation result of sudden climbing under smooth conditions of the travel drive system with PID control strategy.
Figure 14(c). Simulation result of sudden climbing under smooth conditions of the travel drive system with fuzzy PID control strategy.
Figure 14(d). Simulation result of sudden climbing under smooth conditions of the travel drive system with sliding mode control strategy.
Figure 15(a). Simulation results of the sudden crossing of the smooth state of the travel drive system with three control strategies.
Figure 15(b). Simulation result of the sudden crossing of the smooth state of the travel drive system with PID control strategy.
Figure 15(c). Simulation result of the sudden crossing of the smooth state of the travel drive system with fuzzy PID control strategy.
Figure 15(d). Simulation result of the sudden crossing of the smooth state of the travel drive system with sliding mode control strategy.
Figure 16(a). Simulation results of sudden load shedding under smooth conditions of the travel drive system with three control strategies.
Figure 16(b). Simulation result of sudden load shedding under smooth conditions of the travel drive system with PID control strategy.
Figure 16(c). Simulation result of sudden load shedding under smooth conditions of the travel drive system with fuzzy PID control strategy.
Figure 16(d). Simulation result of sudden load shedding under smooth conditions of the travel drive system with sliding mode control strategy.
Please see the attachment

Reviewer 2 Report
1- Replace the expression “Fuzzy PID” with “adaptive Fuzzy PID”
The first expression makes confusion maybe means you use Fuzzy to adapt PID constants or you used a fuzzy like PID controller. But the second expression means that the PID parameters are adjusted by fuzzy rules in real-time.
2- Explain why you did not use a symmetrical diagonal to construct the rule base table or refer to a reference
3- Explain why you did not use a symmetrical fuzzy set in the membership or refer to a reference
4- Explain why you use a different fuzzy set for PB and NB
5- In Figure 13 legend, replace “initial” with “a set point” or “reference”.
6- In Abstract
“The test 17 results revealed that when the current value of the leafy vegetable harvester travel speed deviated 18 from the set value by more than 2%, the dynamic response performance and stability of the DC 19 brushless motor travel drive system based on the sliding mode control strategy was significantly 20 better than that of the PID and fuzzy PID control strategies, and its anti-disturbance was stronger, 21 achieving the function of automatic control of the harvester travel speed.”
And in section 3-Results
“As a result, the travel drive system under the sliding mode control was extremely 386 insensitive to disturbance and more stable than PID and fuzzy PID control when the load 387 was suddenly reduced in the smooth running state of the travel motor, but it oscillated 388 weakly in the secondary steady state range.”
But the results show Sliding mode controller results very bad in steady state, you should illustrate that and discuss with reasons. I think you may improve the results by applying the concept of Multi degree of freedom controller.
7- Verify your simulation results with another previous work
Author Response
Response to Reviewer 2 Comments
Point 1: Replace the expression “Fuzzy PID” with “adaptive Fuzzy PID”.
Response 1: Thank you for your valuable suggestion. The precedent version of the “Fuzzy PID” has been replaced, becoming the “adaptive Fuzzy PID”.
Point 2: Explain why you did not use a symmetrical diagonal to construct the rule base table or refer to a reference.
Response 2: Thank you for your valuable suggestion. In the article, I have derived the fuzzy control rule tables for Kp and Ki based on practical engineering application experience, and the resulting fuzzy control rule tables are not entirely symmetrical about the diagonal.
Point 3: Explain why you did not use a symmetrical fuzzy set in the membership or refer to a reference.
Response 3: Thank you for your valuable suggestion. The fuzzy sets I use in my membership are all {NB, NM, NS, ZO, PS, PM, PB}, which is a left-right, negative-positive symmetric fuzzy set, with NB symmetric to PB, NM symmetric to PM, and NS symmetric to PS.
Point 4: Explain why you use a different fuzzy set for PB and NB.
Response 4: Thank you for your valuable suggestion. PB denotes the bigger positive direction and uses a Gaussian-type subordinate function with a theoretical domain of [-0.8,3]; NB denotes the bigger negative direction and uses a Gaussian-type subordinate function with a theoretical domain of [-3,0.8], so the fuzzy domains of PB and NB are not the same.
Point 5: In Figure 13 legend, replace “initial” with “a set point” or “reference”.
Response 5: Thank you for your valuable suggestion. The precedent version of the “initial” has been replaced, becoming the “a set point”.
Point 6: In Abstract
“The test 17 results revealed that when the current value of the leafy vegetable harvester travel speed deviated 18 from the set value by more than 2%, the dynamic response performance and stability of the DC 19 brushless motor travel drive system based on the sliding mode control strategy was significantly 20 better than that of the PID and fuzzy PID control strategies, and its anti-disturbance was stronger, 21 achieving the function of automatic control of the harvester travel speed.”
And in section 3-Results
“As a result, the travel drive system under the sliding mode control was extremely 386 insensitive to disturbance and more stable than PID and fuzzy PID control when the load 387 was suddenly reduced in the smooth running state of the travel motor, but it oscillated 388 weakly in the secondary steady state range.”
But the results show Sliding mode controller results very bad in steady state, you should illustrate that and discuss with reasons. I think you may improve the results by applying the concept of Multi degree of freedom controller.
Response 6: Thank you for your valuable suggestion. I have consulted the literature on automatic control theory, according to the relevant knowledge of the principle of automatic control, when the travel speed first enters into the ±2% range of the set value and no longer exceeds the time for regulation. The ±2% range of the set value is called steady state. As long as the travel speed is always in the ±2% range of the set value, the system is always in steady state. The travel speed automatic control system under the sliding mode control strategy has a slight oscillation after entering the steady state, but it ensures that the travel speed is always within the ±2% range, so the system is still always in the steady state.
Point 7: Verify your simulation results with another previous work
Response 7: Thank you for your valuable suggestion. In the article, I used field tests to verify the correctness of the simulation tests. Both field and simulation experiments show that the dynamic response performance and stability of the sliding-mode control strategy-based travel speed automatic control system is significantly better than the PID and adaptive fuzzy PID control strategies when the current value of the leafy vegetable harvester travel speed deviates from the set value by more than 2%, and that it is more resistant to perturbations.
Please see the attachment

Round 2
Reviewer 2 Report
The authors considered almost the whole comments